# Machine Learning Models for Prediction of Sex Based on Lumbar Vertebral Morphometry

**DOI:** 10.3390/diagnostics13243630

**Published:** 2023-12-08

**Authors:** Madalina Maria Diac, Gina Madalina Toma, Simona Irina Damian, Marin Fotache, Nicolae Romanov, Daniel Tabian, Gabriela Sechel, Andrei Scripcaru, Monica Hancianu, Diana Bulgaru Iliescu

**Affiliations:** 1Forensic Medicine Sciences Department, Institute of Legal Medicine, University of Medicine and Pharmacy “Grigore T. Popa”, 700115 Iasi, Romania; madalina-maria.diac@umfiasi.ro (M.M.D.); bulgarudiana@yahoo.com (D.B.I.); 2Forensic Medicine Department, “Sf. Ioan” Hospital Suceava, University of Medicine and Pharmacy “Grigore T. Popa”, 700115 Iasi, Romania; 3Alexandru Ioan Cuza University, 700506 Iasi, Romania; fotache@uaic.ro (M.F.); nicolai.romanov3@gmail.com (N.R.); 4Department of Fundamental, Prophylactic and Clinical Disciplines, Medicine Faculty, Transilvania University of Brasov, 500019 Brasov, Romania; daniel.tabian@unitbv.ro (D.T.); gabisechel@yahoo.com (G.S.); 5Forensic Medicine Sciences Department, University of Medicine and Pharmacy “Grigore T. Popa”, 700115 Iasi, Romania; scripcaruand@gmail.com; 6Pharmacy Department, University of Medicine and Pharmacy “Grigore T. Popa”, 700115 Iasi, Romania; monica.hancianu@umfiasi.ro

**Keywords:** forensic identification, machine learning, sex identification, lumbar vertebral column

## Abstract

Background: Identifying skeletal remains has been and will remain a challenge for forensic experts and forensic anthropologists, especially in disasters with multiple victims or skeletal remains in an advanced stage of decomposition. This study examined the performance of two machine learning (ML) algorithms in predicting the person’s sex based only on the morphometry of L1–L5 lumbar vertebrae collected recently from Romanian individuals. The purpose of the present study was to assess whether by using the machine learning (ML) techniques one can obtain a reliable prediction of sex in forensic identification based only on the parameters obtained from the metric analysis of the lumbar spine. Method: This paper built and tuned predictive models with two of the most popular techniques for classification, RF (random forest) and XGB (xgboost). Both series of models used cross-validation and a grid search to find the best combination of hyper-parameters. The best models were selected based on the ROC_AUC (area under curve) metric. Results: The L1–L5 lumbar vertebrae exhibit sexual dimorphism and can be used as predictors in sex prediction. Out of the eight significant predictors for sex, six were found to be particularly important for the RF model, while only three were determined to be important by the XGB model. Conclusions: Even if the data set was small (149 observations), both RF and XGB techniques reliably predicted a person’s sex based only on the L1–L5 measurements. This can prove valuable, especially when only skeletal remains are available. With minor adjustments, the presented ML setup can be transformed into an interactive web service, freely accessible to forensic anthropologists, in which, after entering the L1–L5 measurements of a body/cadaver, they can predict the person’s sex.

## 1. Introduction

Identifying skeletal remains has been and will remain a challenge for forensic experts and forensic anthropologists, especially in disasters with multiple victims or skeletal remains in an advanced stage of decomposition. In such contexts, forensic experts must use knowledge from the field of forensic anthropology, a field which takes interest in the systematic examination of human bones. In order to identify the bones presented for examination as accurately as possible, a first step is to build the biological profile, which involves establishing ethnicity, sex, stature, and age [1,2]. 

Sex determination is a fundamental step in estimating the biological profile from the examination of skeletal remains in forensic anthropology. Most human bones have been used to create various methods of predicting sex. Among the human bones, the coxal bone and the skull are the most accurate for predicting sex, and the method used is a simple macroscopic analysis. There are, however, multiple situations in which these skeletal elements are not available, with the expert being forced to deal with bone fragments or sometimes only with different bones of the human skeleton [3,4,5]. In such circumstances, it is important to develop alternative methods that use other skeletal elements to predict sex. 

The literature mentions only a few studies on the involvement of the spine in developing methods for sex prediction. The spine is a part of the human skeleton used in forensic identification, primarily because of its ability to resist mechanical forces, as well as due to the sexual dysmorphism based on the size and shape of certain vertebrae [6]. Regarding the use of the lumbar spine to sex prediction, the literature mentions several studies on the development of discriminatory functions involving only the L1 and L5 lumbar vertebrae [7,8,9]. The vertebral column represents an important structure of the human skeleton, being involved in multiple daily physical activities, providing the ability to carry various loads. Therefore, the vertebral column has multiple functions and is a complex anatomical structure. It is well known that as a person gets older, the spine undergoes degenerative alterations (osteoporotic and osteoarthritis processes). All these changes lead to morpho-pathological alterations of the spine, translated into a reduced size of the vertebral bodies [10]. 

Data regarding the vertebral morphometry can be obtained using cadavers, bone collection, or by using advanced imaging techniques. The advanced imaging techniques include lateral X-rays, computed tomography, and magnetic resonance imaging. Usually, cadavers and bone collection are the main source of data in forensic research in general and forensic anthropology in particular, but not all the countries have bone collection, while the use of cadavers to evaluate the vertebral morphometry requires some challenging dissection techniques of the vertebral column, which is not impossible but is more difficult and time consuming. The presence of virtual autopsy in some countries makes cadaver research easier by using advanced imaging techniques instead of invasive dissection. As mentioned before, the use of different imagistic scans on living people are the most common and utilized techniques in acquiring data for studies in forensic anthropology. Because MRI-based vertebral morphometry was reported to be more accurate than lateral X-ray-based morphometry, in the present study we used magnetic resonance images to assess sex using machine learning models [8,11,12,13]. Given data recently collected from 149 Romanian adults, the main purpose of this paper was to assess whether the person’s sex can be reliably predicted based only on the metric analysis of the lumbar spine (L1–L5). The predictive models were built using ML techniques which incorporate methods to avoid over-fitting (e.g., cross-validations), data leakage, and provide a good tradeoff between bias and variance. The presented ML setup can further be transformed into an interactive web service, freely accessible to forensic anthropologists, which might also contribute to the extension of the data set by including measurements for individuals from other geographical areas. The interactive web service will be extended, based on this data set, along with sex prediction for age estimation too; this process will show the applicability of machine learning regarding the age estimation for these metric measurements of the lumbar vertebral column for the Romanian population. 

## 2. Materials and Methods

### 2.1. Selection of the Study Lot, Criteria for Inclusion and Exclusion

This study proposes a machine learning method to determine sex starting from morphometric analysis of L1–L5 lumbar vertebrae in a modern Romanian population. A total of 745 lumbar vertebrae (L1–L5) from 149 Romanian individuals (56 men and 93 women) were analyzed by means of MR (magnetic resonance) images in the incidence of T1-FSE (fast spin-echo) of the lumbar vertebral spine. The imaging scans were performed in a Medical Imaging Laboratory in a limited territory in the central region of Romania, with the full consent of the patients according to the working methodology of the Laboratory. The type of study was retrospective. 

The study was conducted in accordance with the Declaration of Helsinki, and the protocol was approved by the Ethics Committee of “Grigore T. Popa” Medicine and Pharmacy University (protocol code 296/30 April 2023). 

The inclusion criteria were age over 17 and the unaltered integrity of the vertebral column. These patients were examined for vertebral pain by neurology and neurosurgery specialists and the MRI scan was recommended to evaluate a possible vertebral pathology as the cause for the pain. The exclusion criteria were represented by cases with advanced scoliotic pathology, traumatic injuries (fractures), or surgery of the lumbar vertebral spine. 

### 2.2. Recording Information in the Database 

For the cases included in the present study, regarding the retained personal data, we noted exclusively the sex and the age of the person to whom the MR scan was performed. 

### 2.3. Working Methodology 

A total number of 230 cases were analyzed, of which 149 MRI images of the lumbar vertebral spine met the criteria for inclusion. 

The present study involved performing three measurements on each of the five lumbar vertebrae, totaling 2235 parameters included in the analysis of sex determination using machine learning methods. 

The measurements performed evaluated the posterior height of the vertebral bodies, the width of the upper and lower plateau of each vertebral body, respectively; the results are presented in Table 1. 

The analysis of MR images and measurements included in the study were performed using the Radiant Dicom Viewer program, by means of the Ruler function (Figure 1). 

### 2.4. Data Analysis and Machine Learning Methodology 

For the 149 MRI images, we collected 15 variables related to L1–L5, as presented in Table 1. Sex also was collected to estimate the performance of subsequent predictive models. Distribution of variables was examined with Exploratory Data Analysis techniques (Figure 2) [14,15]. In machine learning (ML) models, the collinearity of the predictors is not such a critical concern as in classical statistical analysis (e.g., linear or logistic regression). Nevertheless, before building ML models, we removed a series of predictors which recorded large correlations with other predictors (Figure 3 and Figure 4).

Given the nature of the data set (L1–L5 measurement could be recorded from cadavers), the variable to be predicted (sex) was binary. Among many classifiers used in ML, in this paper, the models were built and refined with Random Forests (RFs) and Extreme Gradient Boosting (XGB), two of the most popular ML algorithms [16,17,18,19,20,21,22]. 

Both algorithms grow ensembles of classification or regression trees [23,24]. By building trees through split-variable randomization, RFs [25] manifest an increased prediction accuracy and a decreased prediction variance [26]. 

Boosting processes “weak” learners (e.g., stumps or one-level trees) iteratively using a gradient learning strategy and thus produces “strong” learners [27]. XGB [28] is a regularized implementation of a gradient boosting framework [29] with good performance in both classification and regression [30]. While RF performs better in variance reduction, XGB excels in bias reduction.

Both RF and XGB have hyper-parameters (or tuning parameters) that cannot be learned directly from the data, but they need to be refined [31]. Since larger numbers of assembled trees do not significantly improve the overall performance [32], in this paper, the ntrees parameter was fixed to 700; only two parameters were tuned for the RF models: *mtry* (number of random attributes used for node splitting) and min_n (minimum number of observations in a node as a requirement to continue the tree splitting).

For the XGB models, six hyper-parameters were tuned: learn_rate (learning rate);loss_reduction (min reduction in the loss function for continuing the tree split);tree_depth (max tree depth);sample_size (random samples size);min_n and mtry (as for RF models).

Following the recommendation in [30], the number of trees was not tuned but fixed at 1000 for all XGB models.

The RF and XGB classification models were tuned by choosing in advance 100 (RF) and 300 (XGB) combinations of values for the selected hyper-parameters using a random grid search [33]. The best combination of hyper-parameters was chosen by the Receiver Operating Characteristic Area Under the Curve (ROC-AUC) metric [34]. 

Data leakage was avoided by splitting at random the initial data set into the training subset (70% of the initial set observations) and the testing subset (30%). Overfitting was reduced by repeated k-fold cross validation [34] of the training subset. 

Both algorithms provide the estimated predictors’ contribution to the outcome variation (the variable importance). Among the variable selection methods for RF [35], the permutation-based method was preferred in this study. The importance of variable k is based on the increase in the prediction error in the test set if the variable k’s values are permuted at random. In RF models, through permutation, all correlated predictors are qualified as important if any one of them is important [26]. Of the three scores which generally provide the variable importance in XGB models—gain, cover, and frequency—the xgboost engine focuses on gain [36].

The main interest of this paper was to build a model which properly predicts the sex of a body based solely on the L1–L5 vertebrae measurements. Despite their excellent predictive power, ML algorithms like RF, XGB, or neural networks are opaque. Starting in 2016, scholars and professionals in many areas (medicine included) required more transparency and interpretability for the ML models [37,38]. Of the techniques for interpretable machine learning [39,40,41], for this paper, we used Variable Importance plots, Partial Dependency Plots, and Accumulated Local Effects Plots, as described in the literature [42,43].

Partial Dependency Plots (PDPs) and Accumulated Local Effects Plots (ALEs) are two explanatory tools used for visualization and interpretation of effects that the analyzed features have on model predictions. The idea behind PDPs is to analyze the behavior of model predictions based on one or two selected features [42]. A partial dependency profile is calculated as the mean of ceteris paribus profiles—which is a technique to show the dependence between prediction and a feature variable at the instance level. The shape of the PDP plot will suggest whether the relationship between the output and predictors is linear, monotonic, or complex [43]. These plots provide a simple method to describe the influence that a selected feature has on the outcome, but they have a major disadvantage when the analyzed features are correlated. 

This issue is solved by an ALE, which essentially is the same function of one or two features, but the key difference is how they handle the influence of other features. PDP plots average the predictions and ALE plots use the difference in predictions and accumulate them.

While both PDPs and ALE plots aim to visualize the impact of features on model predictions, ALE plots often provide a more accurate depiction, thus they are the way to go when choosing between these two options [43].

Data were imported, prepared, explored, and analyzed using R version 4.3.0 [44], mainly with the tidyverse ecosystem of packages (dplyr, tidyr, ggplot2, etc.) [45]. Descriptive statistics (Table 2) were generated with the gtsummary package [46], and ggplot2 package was the main tool for the graphics. 

The tidymodels ecosystem of packages (rsample, recipes, parsnip, yardstick, tune, dials, workflows) [47,48,49] was employed for model building and tuning (Figure 5 and Figure 6 were generated with tidymodels and ggplot2 packages). RF models were fitted with the ranger engine [50], whereas the engine used for building XGB models was xgboost [30]. 

The model interpretation (Figure 7, Figure 8, Figure 9, Figure 10, Figure 11 and Figure 12 included) relied on the *DALEX* ecosystem [51], mainly the *ingredients* package [52].

## 3. Results

This section starts with data exploration, by examining the data distribution and correlation among predictors. Subsequently, some details on model building, assessment, and tuning are provided. Finally, models which recorded the best performance are analyzed using variable importance and some other techniques related to model interpretation (explainable AI).

### 3.1. Data Distribution Correlation among Predictors 

Table 2 shows the descriptive statistics for each numerical variable in the data set—the minimal value, the 1st quartile (Q1 or the 25th percentile), the 3rd quartile (Q3 or the 75th percentile), the median (the 50th percentile), and the maximal value. The average value (mean) is accompanied by the standard deviation (SD). 

As the main interest of this paper was to build models for sex prediction based on measurements of the L1–L5 vertebrae, Figure 2 displays the distribution of numeric variables by sex. Despite some differences in between sexes, the shape of the distribution is generally similar, with males’ measurements appearing to exceed the values for females. Nevertheless, here, we were not interested in the analysis of the statistical differences between sexes for the L1–L5 variables. 

Before building the ML models, predictors’ collinearity was assessed and fixed. Figure 3 shows the correlation matrix among all numeric variables in the initial data set. 

Classical statistical techniques, such as linear and logistic regression, require removing large correlations among predictors, since collinearity usually affects model performance. Even if both RF and XGB models handle collinearity much better, we removed predictors recording correlation coefficients larger than 0.75. The final data set contains predictors in Figure 4.

Also, in Figure 4, variable age was removed, since when sex is unknown, a person’s age could also not be determined. Consequently, the final data set on which the ML models were built and tuned contains sex (as the outcome variable) and all variables in Figure 4 (as predictors).

### 3.2. Model Building and Refinement

The 149-observation data set was randomly split into the training data set which contained 111 records (about 75%) and the testing data set containing 38 records (25%). All further model training, tuning, and selection were performed only on the training data set. The testing data set was used solely for estimating the model performance on new data (data not “seen” during the training steps). This is a basic prerequisite in ML model building.

To reduce overfitting, the training subset was further split randomly into five cross-validation folds. In each training fold, the data was subsequently split into the analysis subset and the validation subset. 

For each cross-validation fold, 100 RF models (each model incorporated 700 trees) were built and assessed for each combination of (mtry, min_n) hyper-parameters extracted through a random grid search. Figure 5 shows the values of the two main performance metrics of classification (accuracy and roc_auc) when mtry (# Randomly Selected Predictors) and min_n (Minimal Node Size) varied within their value range extracted through a grid search.

Figure 5 shows that, for both hyper-parameters, larger values generally decrease the model performance, and the best values of both accuracy and roc_auc for the training data set were recorded during the first half of hyper-parameters’ range. The best models were chosen using the roc_auc metric. For RF, the best performance along the five cross-validation folds was recorded for mtry = 1 and min = 8.

XGB models were built using the same subsets/folds as for RF. But as the number of hyper-parameters to be tuned was three times higher than in RF models, for the XGB models, 300 combinations of the hyper-parameter value were selected through a random grid search (each model incorporated 1000 trees). One of the remarkable features of the tidymodels ecosystem is that the packages managing the grid search (tune and dial) automatically extract the appropriate values of the hyper-parameters, according to the data set characteristics, without any tweaking from the user. This is useful especially for the XGB hyper-parameters such as learning rate, loss reduction, and sample size.

Figure 6 displays the values of accuracy and roc_auc when varying the XGB hyper-parameters.

For XGB models, the best roc_auc (averaged along the cross-validation folds) was recorded for the following combination: mtry = 2, min_n = 7, learn_rate = 0.0002989344, loss_reduction = 0.0000000001035262, tree_depth = 9, and sample_size = 0.9337572.

The “moment of truth” for the predictive models is how they perform on new (“unseen”) data. There are models which confound the pattern with the noise, i.e., they found non-existing patterns in data (overfitting). This is the role of the testing subset. After identifying the best combination of hyper-parameters, the best RF and XGB models were applied for the testing data. Table 3 displays both accuracy and roc_auc performance metrics for the selected/best RF and XGB models. 

Selected models recorded good performance on both metrics. While in terms of accuracy, the XGB selected model overperformed the RF selected model (0.816 vs. 0.789), when considering the roc_auc, RF performed better (0.963 vs. 0.868). To summarize, in terms of prediction performance, both RF and XGB selected models seem to supply good predictions of the person’s sex based on her/his L1–L5 vertebrae measurements. 

### 3.3. Model Interpretation 

After assessing the predictive power of the ML models built upon RF and XGB, next, we were interested in exploring the predictors’ importance in the models and how the most important predictors were associated with the outcome (sex) within each selected model. Figure 7 mirrors the predictors’ importance for the RF (left) and XGB models (right), as estimated by the ingredients package. 

Out of eight predictors, six were found to be particularly important for the RF model, while only three were determined to be important by the XGB model. For RF, height_l4 emerged as the most important feature, followed by width_sup_l1 and width_sup_l5 ranking second and third, respectively. These two features were also identified as the most important by the XGB model, with width_sup_l5 being the most important variable and width_sup_l1 being the second (most) important. The top 3 for XGB was completed by height_l1, which, intriguingly, was the least important in the RF model. The other three variables that were qualified as important by the RF model were width_sup_l2, height_l3, and height_l5, filling the fourth, fifth, and sixth positions, respectively.

For the PDP and ALE plot analysis (Figure 8, Figure 9, Figure 10, Figure 11 and Figure 12), only the variable importance as estimated by the RF selected model was considered, since its roc_auc metric was the highest on the test set. From the variable importance plot in the left side of Figure 7, the top 5 most important predictors were examined. For each top predictor, the figure includes three charts: the PDP plot, the ALE plot (for checking if the PDP plot is affected by correlation with other predictors), and the density curve (to identify the ranges where models were fitted on a small number of predictor values and thus the interpretation needs extra precaution).

Both the PDP (Figure 8—left) and ALE (Figure 8—center) plots for the height_l4 variable suggest that the probability of sex being predicted as “female” drops after a value of 2.5, i.e., values of height_l4 larger than 2.5 are more likely to be associated with males. The rather weird jumps on the left and right side of the plots can be explained by the low number of values in those regions, as can be seen in the density curve for the variable in Figure 8 (right).

The PDP plot for the second most important feature, width_sup_l1, presented in Figure 9 (left), and the ALE plot (center) follow a similar pattern. Both of them suggest that the probability outcome of sex being “female” is higher while the values for width_sup_l1 are low and slowly decreases as the values rise, especially after 3.6.

For the variable *width_sup_l5,* both the PDP plot (Figure 10—left) and ALE plot (Figure 10—center) exhibit similar results. A value below 4.2 is strongly associated with a high probability of the value “female” for the outcome. Notably, the value is also the starting point of a steep decrease in the probability of a person being a female. The slight increase after 4.5 can, once again, be explained by the low number of observations in that range, as seen in Figure 10—right.

The *width_sup_l2* feature is associated with a higher probability of sex being female for values less than 3.6, and the probability starts to lower for values up to 4.6, as can be seen in Figure 11 (left and middle). Marginal intervals contain outliers which result in steep increases or decreases in the plot curves.

Finally, as seen in Figure 12 (left and center), the height_l3 feature presents a descending curve, meaning that values lower than 2.5 are associated with a higher probability of the outcome being “female”, and values greater than 2.5 decrease the chance of the “sex” being predicted as “female”, but here, the outcome probability descends in a more gradual manner.

Generally, larger values of L1–L5 vertebrae measurements are associated with males.

## 4. Discussion

While artificial intelligence (AI) can be considered an area of research aimed at mimicking human abilities, machine learning is a specific subset of AI that develops a computer’s ability to learn. The interest in ML is due to various factors such as the increasing volume and variety of data available on the internet, cheaper and more powerful computer processing, and affordable data storage. Advances in ML have led to the development of the ability to quickly and automatically produce models that can analyze a significant amount of complex data with faster and more accurate results. 

The purpose of the present study was to observe whether by using the ML method, there is a good predictability of sex in forensic identification based on parameters obtained from the metric analysis of the lumbar spine specific to the Romanian population. 

Generating a sex prediction model is based on solving a classification task. Classification is one of the most commonly used exploratory tasks in ML [53]. 

In this regard, we used MR images, due to their reliability and performance in visualizing the spine, focusing on the lumbar spine, taking into account parameters such as the height and width of the upper and lower plateau of each lumbar vertebra L1–L5. Of all the measurements performed, only the heights of the L1–L5 vertebrae, respectively, the dimensions of the upper plateaus of the first two vertebrae, L1 and L2, and of the fifth lumbar vertebra, L5, were included in the present study. They were shown to meet all the characteristics to be entered into the ML classification. 

In forensic medicine, and especially in forensic identification, the daily practice of providing correct and complete answers both for justice and for humanitarian and ethical reasons leads to the need of developing and creating as many methods as possible adapted to the new living conditions. Thus, the involvement of machine learning techniques in determining certain parameters that create the biological profile of an individual, in this case determining sex, is a primary necessity of research in this field. 

Sexual dimorphism can be represented on almost every bone component in the cranial and postcranial skeleton. In the present study, we chose to highlight the sexual dimorphism provided by the lumbar spine, describing differences in its morphometry between males and females and generating a machine learning model to accurately predict a person’s sex. According to the results, it is observed that the measurements under discussion show higher values in males compared to females for the Romanian population, which also follows the results from specific studies of other population groups.

Sexual dimorphism of vertebrae is fundamentally based on size, with male individuals generally being larger than female individuals. Previous studies have shown different results on statistically significant differences between sexes in vertebral regions, but all identify dimorphism in vertebral body measurements [8,54,55,56,57,58,59]. Studies led by Taylor and Twomey [54] suggest that these differences may be due to differential growth rates between males and females during puberty, early growth of vertebrae in female individuals, and a greater increase in width in male individuals. In addition, bone size, shape, and density are also influenced by physical activity and mechanical stress [60]. The smaller size of the vertebral body in female individuals is associated with greater flexibility of the spine compared to an accentuated lumbar lordosis in response to the biomechanical needs of pregnancy [8,61,62].

The present study proposes a machine learning model in which, for both sexes, the selected models performed well, in terms of predictive performance, and both selected RF and XGB models appear to predict the person’s sex based on L1–L5 measurements. In a modern African population study [8], significant sex differences were identified in several metric traits of the lumbar vertebrae, and the multiple discriminating functions generated from the analyzed data were able to predict sex with satisfactory accuracy. Other studies using individual postcranial elements such as the femur [63,64,65], tibia [63,66], patella [67,68], humerus [69,70], radius and ulna [71], and various hand and foot bones [72] showed comparable performance to the present study.

This paper offers two models of ML, RF, and XGB, each with its own characteristics, and presenting different performance, random forest having the best. For both, we used two metrics (accuracy and roc_auc), the latter being the most used to highlight model performance.

For both metrics, the selected models recorded good performance. While in terms of accuracy the XGB selected model overperforms the RF selected models (0.816 vs. 0.789), when considering the roc_auc, RF performed better (0.963 vs. 0.868). To summarize, in terms of prediction performance, both RF and XGB selected models seem to predict the person’s sex based on the L1–L5 measurements.

Because the identity of individuals must be predicted quickly and accurately in events such as war, natural disasters, or fires, which profoundly affect society, imaging (virtual forensic) scanning of cadaveric bodies and MLs used in the present study show that prediction time can be minimized and high accuracy can be achieved depending on the situation. Given the high accuracy and reliability of results for both RF and XGB algorithms, it is believed that this study will strengthen and contribute to studies related to sex prediction and its associations with L1–L5 measurements.

### Limits of the Study

Even though with the current data set, the ML models recorded good results, the number of observations is low by ML standards and needs to be enlarged in future studies.

RF and XGB findings should also be compared with results provided not only by other statistical techniques, such as logistic regression and discriminant analysis, but also with other ML classifiers, such as Support Vector Machines, Naïve Bayes, or Neural Networks (admittedly, some algorithms will require much larger data sets for training).

As the training data set is small and homogeneous, application of the current ML setup should be carried out with extra caution for people originating in other geographic regions.

Age may prove to mediate the association between L1–L5 measurements and sex, and this should be addressed by further research.

## 5. Conclusions

Data collection and data analysis methods and tools presented in this paper provide reliable information and results with large applicability in the future for sex prediction based on vertebral column measurements for the Romanian adult population. This is even more important when only skeleton parts are available for anthropological analysis. Future research may consider more measurements, describing larger segments of the vertebral column, as extracted from CT and MR images. Also, future research will be carried out regarding the age estimation based on this data set; this aspect will be important to show the importance of and the impact of vertebral column on age estimation.

The present work could serve as a good starting point in the introduction and development of machine learning models in Romanian forensic anthropology. Based on the setup deployed for this study, a digital interface may be implemented and made available to all practitioners of the forensic network in Romania. This interface could be developed by including additional parameters for supporting forensic identification, such as parts of the biological profile, postmortem interval, etc. Such an interface would contribute to forensic medicine in Romania.

## Figures and Tables

**Figure 1 diagnostics-13-03630-f001:**
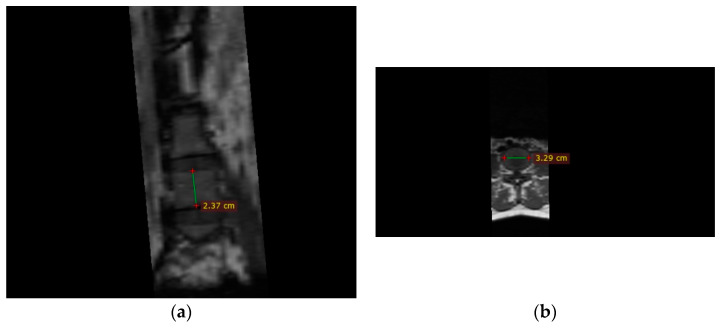
Measurements on vertebral column (exemplification of vertebral body height (**a**) and width of superior endplate (**b**)).

**Figure 2 diagnostics-13-03630-f002:**
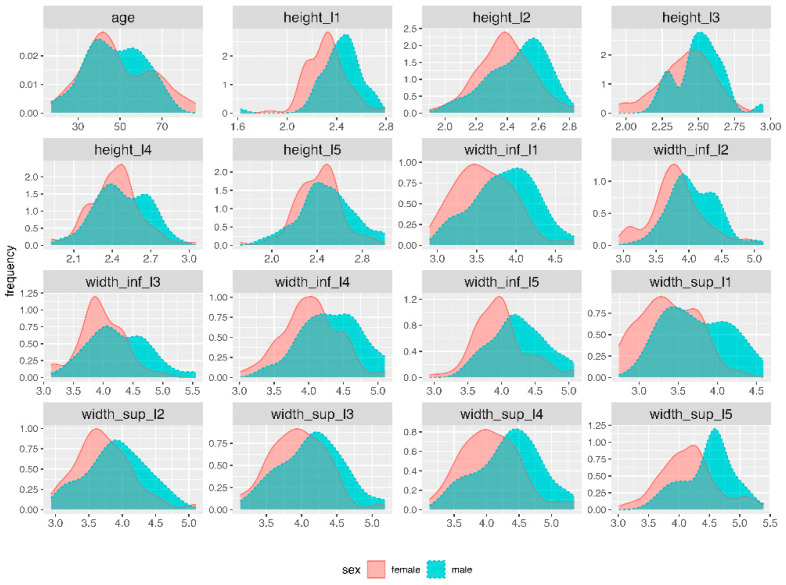
Distribution of numeric variables, grouped by sex.

**Figure 3 diagnostics-13-03630-f003:**
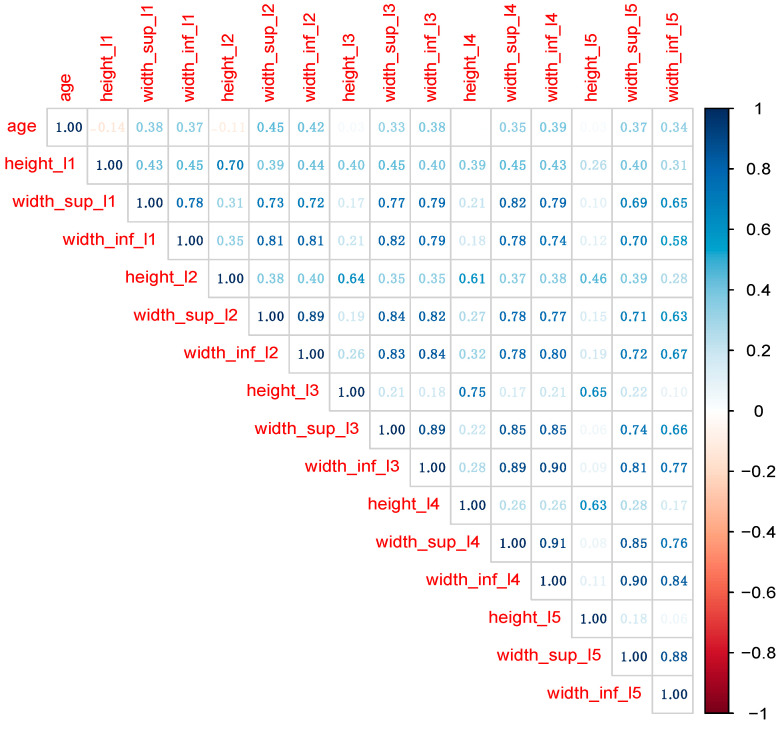
The correlation plot among numerical variables in the initial data set.

**Figure 4 diagnostics-13-03630-f004:**
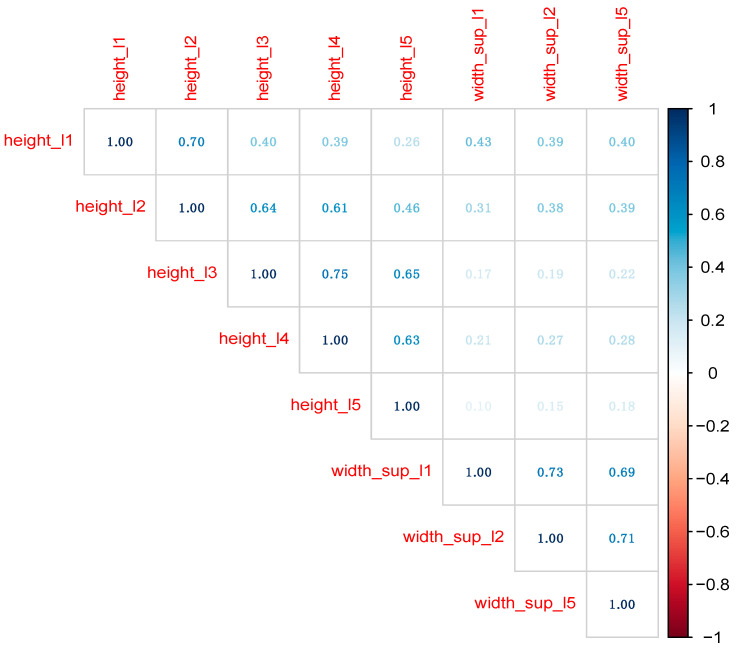
The correlation plot among numeric variables in the final data set.

**Figure 5 diagnostics-13-03630-f005:**
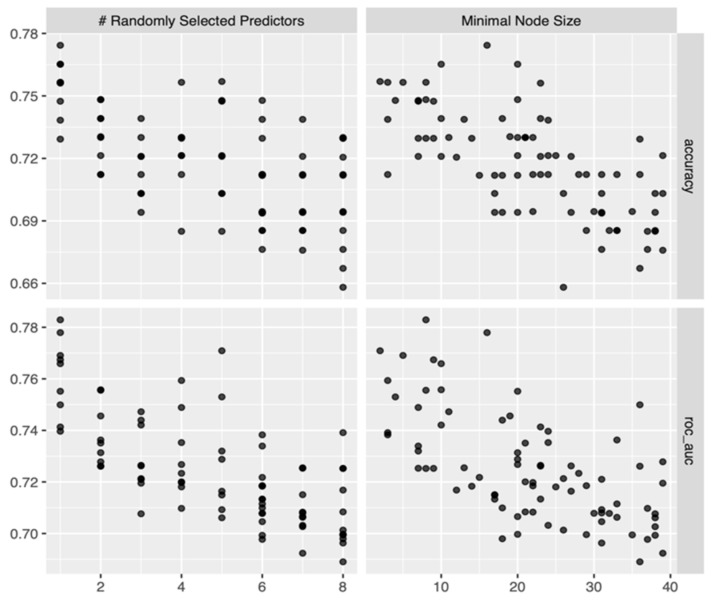
Performance metrics vs. hyper-parameter values for the trained RF models.

**Figure 6 diagnostics-13-03630-f006:**
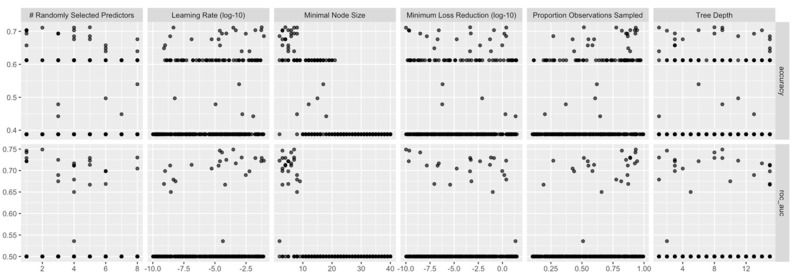
Performance metrics vs. hyper-parameter values for the trained XGB models.

**Figure 7 diagnostics-13-03630-f007:**
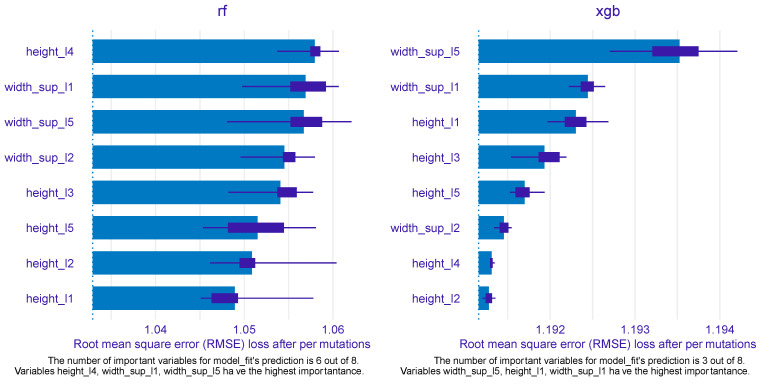
Variable importance for the best random forest and xgboost models, as estimated by the ingredients package.

**Figure 8 diagnostics-13-03630-f008:**
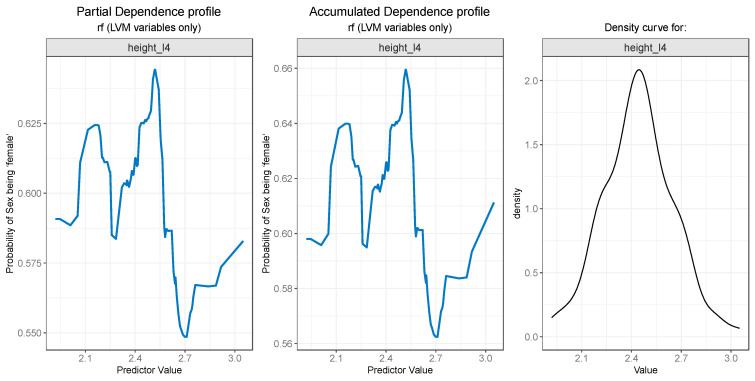
Partial Dependence and Accumulated profiles for the most important predictor in RF.

**Figure 9 diagnostics-13-03630-f009:**
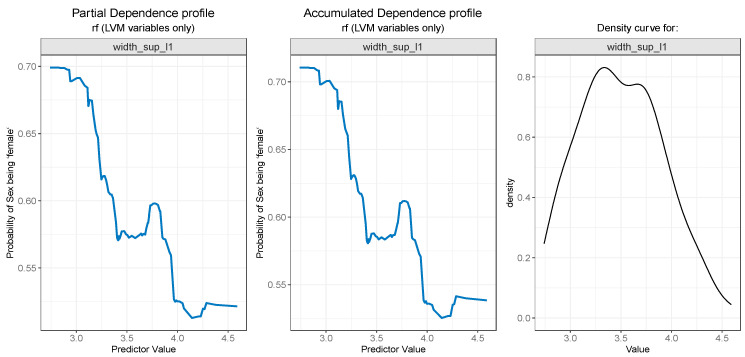
Partial Dependence and Accumulated profiles for the 2nd most important predictor in RF.

**Figure 10 diagnostics-13-03630-f010:**
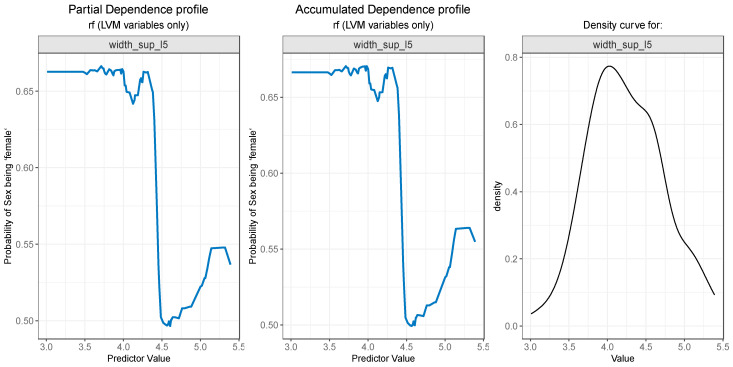
Partial Dependence and Accumulated profiles for the 3rd most important predictor in RF.

**Figure 11 diagnostics-13-03630-f011:**
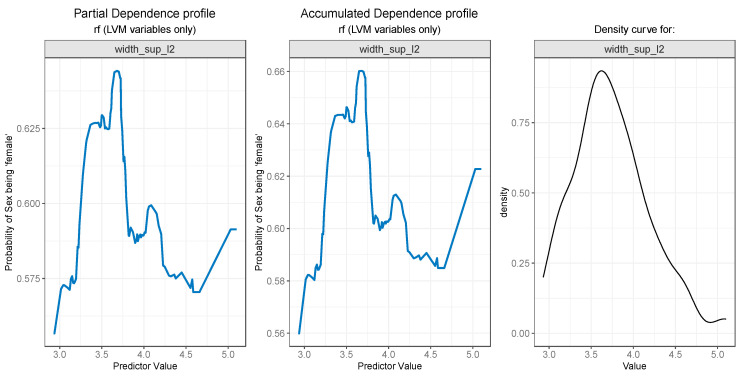
Partial Dependence and Accumulated profiles for the 4th most important predictor in RF.

**Figure 12 diagnostics-13-03630-f012:**
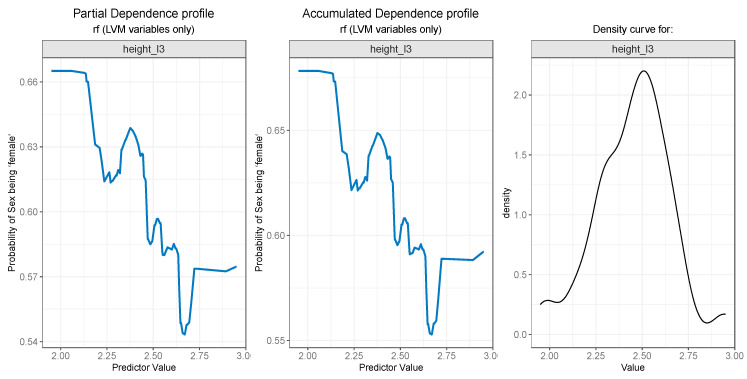
Partial Dependence and Accumulated profiles for the 5th most important predictor in RF.

**Table 1 diagnostics-13-03630-t001:** Measurement of the vertebral column L1–L5.

Measurement	Abbreviation	Vertebrae	Definition
Width of superior endplate	Width_sup_lx	L1–L5	Distance between the most lateral edges of the superior plate of the vertebrae
Width of inferior endplate	Width_inf_lx	L1–L5	Distance between the most lateral edges of the inferior plate of the vertebrae
Posterior height of the vertebral body	Heigth_lx	L1–L5	Posterior height of the vertebral body from the left bisecting plane at the posterior part of the vertebral body at the point which can get the largest height

**Table 2 diagnostics-13-03630-t002:** Descriptive statistics for numerical variables.

Variable	Min	Q1	Median	Q3	Max	Mean	SD
**age**	17	38	46	60	86	48	15
**height_l1**	1.62	2.26	2.36	2.47	2.79	2.36	0.17
**width_sup_l1**	2.74	3.22	3.47	3.78	4.59	3.52	0.43
**width_inf_l1**	2.90	3.41	3.70	3.98	4.74	3.69	0.40
**height_l2**	1.90	2.30	2.41	2.55	2.83	2.42	0.18
**width_sup_l2**	2.93	3.49	3.77	4.05	5.10	3.80	0.44
**width_inf_l2**	2.93	3.67	3.86	4.18	5.14	3.91	0.40
**height_l3**	1.95	2.32	2.47	2.56	2.95	2.45	0.19
**width_sup_l3**	3.13	3.73	4.02	4.32	5.18	4.03	0.43
**width_inf_l3**	3.12	3.79	4.05	4.34	5.55	4.08	0.45
**height_l4**	1.92	2.32	2.44	2.57	3.05	2.44	0.20
**width_sup_l4**	3.12	3.82	4.19	4.48	5.35	4.17	0.48
**width_inf_l4**	3.01	3.86	4.12	4.49	5.10	4.14	0.44
**height_l5**	1.72	2.31	2.44	2.56	3.00	2.44	0.22
**width_sup_l5**	3.01	3.96	4.27	4.59	5.39	4.28	0.47
**width_inf_l5**	2.93	3.80	4.05	4.41	5.08	4.09	0.42

**Table 3 diagnostics-13-03630-t003:** Model performance on new data (the test data subset).

Algorithm	Metric	Estimate
**rf**	accuracy	0.78947
**xgb**	accuracy	0.81579
**rf**	roc_auc	0.96308
**xgb**	roc_auc	0.86770

## Data Availability

Data is contained within the article.

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
