# Peer review of "Machine Learning Models for Prediction of Sex Based on Lumbar Vertebral Morphometry"

_diagnostics, 2023, doi:10.3390/diagnostics13243630_

Round 1

Reviewer 1 Report

Comments and Suggestions for Authors

Based on the review, I am suggesting following comments; 

1.     This paper not shown how the ML methods predicts gender?. It stated only about comparison of ML methods using various parameters.

2.     Objective of the work is not covered in abstract. In addition, what is the novelty used in this paper?. Give these details in abstract.

3.     The introduction section is not clear. What is the motivation? Why this area is critical and important? What are the contributions? These points are supposed to be covered in the introduction section

4.     Section 2. Materials and Methods is too weak. It would cover architecture of proposed work, flow of implementation and novel method implemented.

5.     How the figures are plotted?. Source of the data are missing. It is difficult to assess the experimental results without input data.

6.     Please mention the software (the infrastructure) you used to implement your proposed approach.

7.     Your paper is not well structured, reorganize it to the journal format, you can find the guidance on the journal website.

8.     Re-write the conclusion section to focus on what have you achieved from the study, contributions of the study to academics and practices, and recommendations of future works. Besides, the conclusion must include open problems and wider future research avenues

Author Response

Dear Reviewer, 

Thank you for your comments, they are valuable for our work. Please find attached our response.

Respectfully

Authors

Reviewer 2 Report

Comments and Suggestions for Authors

Introduction

The introduction is very poorly written. Most of the sentences are not cited with references.

For example, the following statement should be cited with relevant references.  

“The literature mentions only a few studies on the involvement of the spine in 57 developing methods for gender estimation”.

The spine is a part of the human skeleton, used 58 in forensic identification, primarily because of its ability to resist mechanical forces, as well as due to the sexual dysmorphism based on the size and shape of certain vertebrae.

The rationale for the study needs to be clarified at the end of the introduction section.

The flow of the problem statement is missing.

Methods

Ethic Committee approval number should be mentioned.

Was the sample size calculation performed to test the study’s hypothesis? Is 149 patient’s data is sufficient?

Study setting: Where the study is conducted? The clinical indications of MRI of these patients need to be mentioned.

Results

The table 2 font is too small, it can be increased for better clarity.

Rest of the results section is well written.

Discussion

The limitations of the study need to be added. Most of the sentences are not supported by any references. It is suggested to cite those sentences. For an example following paragraph, there is no supportive evidence.

In forensic medicine, and especially in forensic identification, the daily practice of providing the most correct answers both for justice and for humanitarian and ethical reasons, leads to the need to develop and create as many new methods as possible adapted to the new living conditions of humanity. Thus, the involvement of machine learning techniques in determining certain parameters that create the biological profile of an 366 individual, in this case determining gender, is a primary necessity in research in this field. Sexual dysmorphism can be represented on almost every bone component in the cranial and postcranial skeleton.

Limitations of the study need to be added.

The conclusion could be more precise.

There are grammar mistakes throughout the manuscript. Language Editing is a need from the Expert.

Comments on the Quality of English Language

There are grammar mistakes throughout the manuscript. Language Editing is a need from the Expert.

Author Response

(The authors gave the same response as above.)

Reviewer 3 Report

Comments and Suggestions for Authors

The topic of biological profiling, in particular the determination of sex, age or stature, is topical and is particularly needed in forensics. This article contains information on sex determination from the lumbar spine using ML models. The article provides a lot of valuable statistical analysis and important results. Despite the many positive aspects offered by the present study, there are some important points that should be revised and interpreted with caution. Please see my comments below:

-            Instead of “forensic doctor” please use "forensic pathologist", "forensic expert/anthropologist" and revise this throughout the manuscript.

- I suggest using the term sex, as it refers to biological characteristics, rather than gender, as it refers to the imprint of social and cultural circumstances. Please revise this throughout the manuscript, including title, abstract and keywords.

- Race should be replaced by “ethnic variety” or ethnicity.

- One sentence “There are however, multiple situations in which these two bones are not available, on the contrary even having to deal with bone fragments or sometimes only with different bones of the human skeleton.” Please do not refer to the cranial bones and pelvic bones as two bones, I understand what you mean, but it should be revised and interpreted correctly.

- In the introduction, lines 56-57 and lines 57-59 - please refer to the specific studies at the end of the sentence.

- Revise – “sexual dysmorphism” to sexual dimorphism.

- The introduction should state the aim of the study and clarify the objective contribution of the present study, please add.

- The use of “man” and “woman” should be changed to “male” and “female”, respectively, to reflect modern standards in forensics and to account for possible non-heteronormative identities among the participants.

- The authors should distinguish whether the previous studies were performed on skeletal remains (direct measurements of the lumbar vertebrae) or whether the studies (measurements) and sex prediction models were performed using different imaging techniques (X-ray, CT, CBTCT, PMCT, MRI) – this should be mentioned in the introduction.

- The word “Tabel 1” is misspelled, correct this. Are the results in Table 1 regardless sex? This should be added in the title.

- I suggest revising Figure 1 - point out with an arrow or the letters a, b which parameter is meant, and also the figure is not very readable, it should be enlarged.

- In the introduction or in the limitations of the study, preferably both, it should be pointed out that the lumbar part of the spine is more prone to various injuries or degenerative changes that depend on physical activity, facts that could have a significant impact on the morphology and measurements of the lumbar spine.

- The age range of the participants should be mentioned in the Material and Method section. Table 1 shows that the age range is between 17 and 86 years. Could age be a factor influencing the assessment of sex? I think this question should be carefully discussed as degenerative changes in the lumbar spine are caused by age – this should be mentioned in the limitations of the study and also in the discussion.

- In Figure 2 it is not clear which curved line is male or female, please revise.

- What does Figure 5 stand for? For all figures or diagrams, authors should provide sufficient information about what the x-axis or y-axis represents.

- The manuscript is missing the paragraph or subsection on the limitations of the study - at least two of the above areas should be included in the proposed paragraph on the limitations of the study.

- In the conclusion, the authors state, “Machine learning is more accurate in determining gender than regression/discriminatory function analysis”." This statement is too bold and presumptuous. Did the authors compare ML models and those of DF or RegA? Overall, the conclusion should be interpreted with caution as several factors, such as the age of the sample and physical activity (which I assume was not part of the scope of the study), could influence the lumbar measurements.

Comments on the Quality of English Language

Minor editing of English language required

Author Response

(The authors gave the same response as above.)

Round 2

Reviewer 2 Report

Comments and Suggestions for Authors

The revised manuscript is improved by considering all the suggested comments. 

There are minor language errors in the manuscript. All of these need to be corrected. 

Some of the figure legends are short, they can be expanded for better understanding to the readers. 

Comments on the Quality of English Language

There are minor grammar errors in the manuscript. 

Author Response

Dear Reviewer, 

Thank you for your valuable suggestions and for your time to revise our manuscript

Respectfully

Madalina Diac

Reviewer 3 Report

Comments and Suggestions for Authors

The authors have revised the paper, incorporated all the comments I made and discussed them very thoroughly. Nevertheless, there are a few minor points that should be considered:

- These words are misspelled: “incoporate, accesible”, please revise.

- I understand that an age estimation study is planned, and you have also briefly mentioned this in the limitations of the study, but I still recommend emphasising this in the introduction when defining the aims of the study and also in the conclusion, as the age range in the present study is very wide.

Comments on the Quality of English Language

Minor editing of English language required

Author Response

(The authors gave the same response as above.)
